# Receptor-Arrestin Interactions: The GPCR Perspective

**DOI:** 10.3390/biom11020218

**Published:** 2021-02-04

**Authors:** Mohammad Seyedabadi, Mehdi Gharghabi, Eugenia V. Gurevich, Vsevolod V. Gurevich

**Affiliations:** 1Department of Toxicology & Pharmacology, Faculty of Pharmacy, Mazandaran University of Medical Sciences, Sari 48471-93698, Iran; m.seyedabadi@mazums.ac.ir; 2Pharmaceutical Sciences Research Center, Faculty of Pharmacy, Mazandaran University of Medical Sciences, Sari 48167-75952, Iran; 3Department of Cancer Biology and Genetics, The Ohio State University Wexner Medical Center, Columbus, OH 43210, USA; mehdi.gharghabi@osumc.edu; 4Department of Pharmacology, Vanderbilt University, Nashville, TN 37232, USA; eugenia.gurevich@vanderbilt.edu

**Keywords:** arrestin, GPCR, protein–protein interactions, signaling, conformational change

## Abstract

Arrestins are a small family of four proteins in most vertebrates that bind hundreds of different G protein-coupled receptors (GPCRs). Arrestin binding to a GPCR has at least three functions: precluding further receptor coupling to G proteins, facilitating receptor internalization, and initiating distinct arrestin-mediated signaling. The molecular mechanism of arrestin–GPCR interactions has been extensively studied and discussed from the “arrestin perspective”, focusing on the roles of arrestin elements in receptor binding. Here, we discuss this phenomenon from the “receptor perspective”, focusing on the receptor elements involved in arrestin binding and emphasizing existing gaps in our knowledge that need to be filled. It is vitally important to understand the role of receptor elements in arrestin activation and how the interaction of each of these elements with arrestin contributes to the latter’s transition to the high-affinity binding state. A more precise knowledge of the molecular mechanisms of arrestin activation is needed to enable the construction of arrestin mutants with desired functional characteristics.

## 1. Introduction

Arrestins are critical players in the homologous desensitization of G protein-coupled receptors (GPCRs). Active GPCRs interact with cognate heterotrimeric G proteins, catalyzing GDP/GTP exchange on their α-subunits. GTP binding to the G protein α-subunit promotes the dissociation of the G protein from the receptor and separation of its α- and βγ-subunits. The classical paradigm of homologous desensitization posits that eventually, the active receptor is phosphorylated by G protein-coupled receptor kinases (GRKs) (reviewed in [1]). Arrestins bind active phosphorylated receptors with high affinity [2]. The receptor binding of G proteins is transient due to the abundance of GTP in the cytoplasm, whereas the binding of arrestins to receptors is not. Thus, after receptor phosphorylation, arrestins outcompete G proteins, shutting down G protein-mediated signaling [3]. The formation of the arrestin-receptor complex also “activates” arrestins, inducing global conformational changes in the arrestin molecule that enable its transition into a state capable of binding the receptor with high affinity. “Active” GPCR-bound arrestins recruit numerous trafficking and signaling proteins [4], promoting receptor internalization and facilitating the signaling in several pathways [5,6]. The realm of arrestin activity goes beyond GPCRs and includes atypical seven transmembrane domain receptors (7TMRs), such as frizzled and smoothened receptors, receptor tyrosine kinases, cytokine receptors, and ion channels [7,8,9]. Quite a few reviews have discussed the role of particular arrestin elements in receptor binding and the consequent signaling [2,5,6,10], but the equally important GPCR side of the story has received a lot less attention.

Free and receptor-bound arrestins are different not only structurally [11] but also functionally [5,12]. Thus, it is important from a biological standpoint to determine what parts of the receptor facilitate arrestin’s transition from one state to the other, the molecular mechanisms whereby individual receptor elements facilitate this transition, and the role of particular interactions between receptor and arrestin residues in this process. Below, we summarize existing data and point out the gaps in current knowledge that need to be filled. We focus on GPCR elements that engage arrestins and, where known, on the actual role of these receptor elements in arrestin binding and its transition into an “active” signaling-competent conformation. We present fine molecular details which might be of interest only to those who work on the structure–function of arrestins and GPCRs. Therefore, we have emphasized the qualitative changes in both arrestins and GPCRs that contribute to the big picture of the regulation of cell signaling, where GPCRs, being the most numerous family of signaling proteins and targeted by about a third of clinically used drugs [13], play a prominent role. While sequence conservation in the GPCR super-family is fairly low [14], all GPCRs have a similar topology: an extracellular N-terminus, seven transmembrane α-helices (TM1-7) connected by three intracellular (ICL1-3) and three extracellular (ECL1-3) loops, and a cytoplasmic C-terminus, the beginning of which, between TM7 and the palmitoylation site, often forms helix 8.

## 2. Where Arrestins Start: Structure in the Basal State

Most vertebrates express four arrestin subtypes: visual arrestin-1 and -4 (We use systematic names of arrestin proteins, where the number after the dash indicates the order of cloning: arrestin-1 (historic names S-antigen, 48 kDa protein, visual or rod arrestin), arrestin-2 (β-arrestin or β-arrestin-1), arrestin-3 (β-arrestin-2 or hTHY-ARRX), and arrestin-4 (cone or X-arrestin)), which are restricted to the photoreceptors in the retina where they quench light-induced signaling of the photopigments in rods and cones, and two ubiquitously expressed non-visual forms, arrestin-2 and -3 (also known as β-arrestin-1 and -2, respectively), which interact with hundreds of different GPCRs [5,15].

Comparison of the crystal structure of bovine arrestin-1 (PDB: 1CF1) [16], bovine arrestin-2 (PDB: 1G4M and 1G4R) [17], bovine arrestin-3 (PDB: 3P2D) [18,19], and tiger salamander (*Ambystoma tigrinum*) arrestin-4 (PDB: 1SUJ) [20] in the basal state reveals the overall similarity of these structures (Figure 1A). All arrestins consist of an N-domain and a C-domain, each formed by a “sandwich” consisting of two layers of β-strands. In addition to the extensive interface where the bodies of the two domains interact, there are two links between the domains: the inter-domain “hinge” and the C-tail. The length of the hinge was shown to be critical for GPCR binding in arrestin-1 [21], as well as non-visual arrestins 2 and 3 [22]. The C-tail makes a loop (not resolved in structures), after which it is anchored to the N-domain via the “three-element interaction” with β-strand I and the only α-helix in arrestins (Figure 1B). Several structural elements in visual arrestin-1 should be noted that might be important for the selectivity of visual arrestins for photopigments, in contrast to the much greater variety of GPCRs with which non-visual arrestins interact [17,20]. Bovine arrestin-1 contains valine in position 90. The large hydrophobic side chain of this valine is localized between the two layers of β-strands and apparently reduces the flexibility of the β-strand sandwich of the N-domain through interactions with several bulky hydrophobic partners [16,17]. Valine in this position is conserved in arrestin-4 (also known as cone arrestin) but is replaced with serine or alanine in non-visual arrestins. While the N-domain of arrestin-4 shares similar H-bonding to that of arrestin-2, its C-domain structure resembles that of arrestin-1, making the structure of arrestin-4 a hybrid of non-visual arrestin-2 and visual arrestin-1. Notably, the loop between β-strands I and II in arrestin-1 contains R18, while the other three arrestins have proline in homologous positions [20]. It has been suggested that this additional positive charge in arrestin-1 ensures its greater preference for phosphorylated over unphosphorylated GPCRs [20]. Indeed, the difference in binding to the phosphorylated and unphosphorylated forms of the same receptor for both non-visual subtypes was experimentally shown to be much less dramatic than for arrestin-1 [23,24,25].

Non-visual arrestins contain proline-rich regions (88–96 and 120–124 in arrestin-2, light magenta in Figure 1B), which are likely responsible for the binding of the SH3 domains of Src family kinases and other proteins containing SH3 domains through polyproline helix II (PPII). Proline-rich regions are absent in both visual arrestins. While the first segment (88–96) is unlikely to produce a PPII-type structure, residues 119–126, located in the connector between α-helix I and the body of the N-domain, can assume a PPII conformation, at least upon arrestin “activation” [27]. However, this requires rearrangement of P121 and N122 upon receptor binding [16,17]. Interestingly, while the connector between the α-helix and the N-domain (Figure 1B) in arrestin-2 has only one PPxP motif, the homologous element in arrestin-3 has two [11,27], which are consensus sequences for binding to the regulatory SH3 domains. In contrast to the other three subtypes, part of the receptor-binding surface in the C-domain of arrestin-3 (Q253-Q262) does not form a contiguous β-sheet via hydrogen bonding (as revealed by the crystal structure at 3.0-Å resolution [18]), allowing increased flexibility of the receptor-binding side of the C-domain, which may explain the more promiscuous nature of arrestin-3, as compared to arrestin-2, in binding to different GPCRs [18].

So far, the highest resolution structure of any arrestin in its basal conformation is that of arrestin-2 (PDB: 1G4M; 1.9 Å) [17]. Therefore, we have used this structure for illustrative purposes (Figure 1B). Like all other subtypes, arrestin-2 contains a polar core (composed of five residues with charged side chains: D26, R169, D290, D297, and R393, green in Figure 1B; expanded in the right insert), which is one of the intra-molecular “clasps” that keep arrestins in their basal conformation. The lariat loop (N281-N311 in arrestin-2, dark blue in Figure 1B), which is highly conserved in all arrestins, contains the main counterion of R169 in the polar core, D290. Mutagenesis data suggest that the salt bridge between these two residues plays a central role in stabilizing the basal state [23,24,28,29,30,31]. The gate loop [16] (part of the lariat loop, D290-N299 in arrestin-2; dark red in Figure 1B) supplies two out of the three negative charges in the polar core. Upon recruitment to the receptor, the polar core is destabilized, and the lariat loop, the C-tail (D383-R408 in arrestin-2, the D383-R393 part resolved in the crystal structure is shown in light blue in Figure 1B), and the N-domain undergo significant structural rearrangements [32,33,34]. The crystal structures of all arrestins in the basal conformation reveal a three-element interaction (Figure 1B, left insert) that involves bulky hydrophobic residues (side chains are shown in yellow in Figure 1B) in the α-helix (98–108 in arrestin-2, orange in Figure 1B) and β-strands I and XX of the N-domain and the C-tail, respectively [16,17,20,26,35]. The interaction of these three elements is disrupted upon binding to the receptor, resulting in structural rearrangement of β-strand I [36,37] and the release of the C-tail [37,38,39]. C-edge loops (dark green in Figure 1B) on the distal tip of the C-domain of arrestin-1 [40] and arrestin-2 [33,34] were found in contact with the detergent or membrane in micelles or lipid nanodiscs, suggesting a role for these residues in the membrane anchoring of receptor-bound arrestins. This element in arrestin-1 was shown to fulfill this function upon rhodopsin binding [41].

## 3. Where Arrestins Go: The Structure of Receptor-Bound Arrestins

Arrestins undergo a global conformational rearrangement upon recruitment to the receptor. First, an interdomain twist, i.e., a rotation of the C-domain relative to the N-domain, is revealed by all structures of “active” arrestins. Interestingly, this twist was predicted long before the structures demonstrated it [42]. The extent of this twist varies between different structures of “active” arrestins [33,34,43,44]. “Active” arrestin conformations fall into two groups, those with small (~8°; PDB: 4ZRG, arrestin-1 R175E [45]; 3UGU, p44 splice variant of arrestin-1 [46]; 6K3F, C7pp-bound arrestin-3 [44]) and large (~18°, PDB: 5TV1, IP_6_-bound-arrestin-3 [27]; 4JQI, V_2_Rpp-bound arrestin-2 [47]; 4J2Q, arrestin-1 p44 [48]; 4ZWJ, rhodopsin-bound arrestin-1 [40]; 5W0P, rhodopsin-bound arrestin-1 [43]; 6UP7, NTS_1_R-bound arrestin-2 [34]) interdomain twists. Specific phosphorylation patterns, i.e., the number and spatial distribution of phosphates, have been suggested as a potential mechanism governing the extent of the interdomain twist [44], but this idea requires experimental testing. In addition to the extent of the interdomain twist, the orientation of arrestin-2 relative to the 7TM bundle of the receptor in complex with M_2_R [33], β_1_AR [49], and NTS_1_R [34] shows a 7°, 20°, and 90° difference, respectively, compared to the rhodopsin-arrestin-1 complex.

Second, the interaction between residues in the polar core is disrupted, as evidenced by the movement of D290 in the lariat loop away from R169, which is an essential contact stabilizing the basal conformation of arrestin-2 [34]. Disruption of the polar core upon activation is a shared phenomenon in the activation of all arrestin subtypes [27,33,34,40,43,44,47,48,49]. Third, the finger, lariat, middle, and C-loops undergo significant rearrangements. While the positions of the lariat and middle loops are similar in all reported receptor/arrestin complexes, the finger loop and C-loop adopt distinct conformations in different structures of “active” arrestins (Figure 2) [33,34,43]. Notably, the finger loop of V_2_Rpp-arrestin-2 complex forms an unstructured region superimposable neither with that of rhodopsin-arrestin-1 nor with that of β_1_AR-arrestin-2. The finger loop of arrestin-2 in complex with β_1_AR forms a β-hairpin, in contrast to the short α-helix in the rhodopsin-arrestin-1 or NTS_1_R-arrestin-2 complexes, and protrudes about 5 Å deeper into the interhelical cavity of the receptor [49].

## 4. Distinct Poses of Receptor-Bound Arrestin

The idea that receptor-bound arrestin does not necessarily have a fixed conformation but can assume different ones was proposed long ago [5]. The simplest explanation of the findings that arrestin binding to the same receptor phosphorylated by different GRKs, presumably at different sites, has distinct signaling consequences [50,51,52] is that the conformation of receptor-bound arrestin depends on the positions of the receptor-attached phosphates, and the actual conformation of bound arrestin determines its signaling capabilities, as was proposed more than a decade ago [53]. Indeed, mutations in the dopamine D_1_ receptor substituting particular phosphorylatable ICL3 residues with alanines or negatively charged phosphomimetics differentially affected its signaling to different protein kinases [54]. Moreover, some arrestin-3 residues significantly change its receptor preference [55,56], even though their homologues in arrestin-1 or -2 do not contact the bound receptors in any of the solved structures. This suggests that these elements participate in the binding, likely in alternative “flavors” of the complexes not resolved in structures. However, all this evidence is indirect. So far, there are very few pieces of direct evidence. First, the same arrestin-2 was found in strikingly different “poses” in complex with different GPCRs (Figure 3) [32,33,34], indicating that there is more than one possible way of arrestin association with the receptor. Second, arrestins are capable of binding to the receptor intracellular core [40] or only to the phosphorylated receptor C-terminus, leaving the core open for the binding of G proteins, forming megaplexes [57]. Third, as far as the complex of a single arrestin bound to a single receptor goes, distance measurements using the pulse electron paramagnetic resonance (EPR) technique called double electron–electron resonance between selected points in rhodopsin and bound arrestin-1 yielded more than one distance between each pair [40,43]. While the most populated distances matched the crystal structure of the complex [40,43], to the delight of crystallographers, the presence of others suggested that the complex can have different “flavors”, only one of which was resolved in the crystal. Two experimental approaches can be used to prove this beyond a reasonable doubt. The first is the elucidation of the structures of arrestin complexes with the same receptor with phosphates in different positions (e.g., using mutant receptors with some of the phosphorylation sites eliminated). The second is the measurement of a sufficient number of distances between certain points in arrestin and in the receptor to develop models of the different flavors of the complex. Receptor and/or arrestin residues engaging the partner in a particular flavor, but not in others, can be mutated. If these mutations affect the observed binding (which is, by definition, the sum of all possible complexes), this would prove that those flavors exist and contribute to the measured interaction.

## 5. GPCR Interhelical Cavity: Role in Arrestin Binding

The classical paradigm posits that arrestin binding to GPCRs requires both receptor activation and phosphorylation, which was directly demonstrated in the case of rhodopsin [58,59] and β_2_-adrenergic receptor (β_2_AR) [60,61]. GPCR activation is accompanied by the movement of transmembrane (TM) helices (the largest movement was observed for TM6), which opens up a cavity on the cytoplasmic side of the receptor [62,63]. This cavity is invariably engaged by the three classes of proteins that preferentially bind active GPCRs: G proteins [64,65,66,67,68,69], GRKs [70,71], and arrestins [32,33,34,37,40]. Available structures of complexes suggest that G_s_ needs a larger cavity than G_i/o_ [64,66,67,68,69,72]. Thus, the questions are (1) whether arrestin needs the same size of cavity as the type of G protein interacting with a particular GPCR; or (2) whether arrestin requires a cavity of a specific size in all GPCRs; and (3) how receptor residues facing this cavity in active GPCRs bind and activate arrestins.

First, let us compare the structure of an inactive and active receptor per se with the structure of an active receptor engaged by a cytoplasmic protein partner. Superimposition of the inactive muscarinic M_2_R (PDB: 3UON) [73] with active M_2_R bound to G_oA_ (PDB: 6OIK) [74] reveals a significant movement of TM6 (K383 ~11.7 Å outward), TM7 (Y440 ~5 Å inward), and H8 (H453 ~4.3 Å upward) at the cytoplasmic face, as well as the movement of TM1 (F21 ~5.1 Å), TM5 (A185 ~5.8 Å), TM6 (F412 ~6.3 Å) and ECL3 (P415 ~5.3 Å) at the extracellular side, upon activation. In addition, TM3, ICL2, TM4, and TM5 at the cytoplasmic side shift by 1.5 to 2.5 Å (Figure 4A–C). However, some of these movements could have been induced by the use of fusion proteins for crystallization, such as the FLAG-tag at the N-terminus and T4-lysozyme between TM5 and TM6. Analysis of the position of residues critical for receptor activation revealed 5-Å movement of Y440 in the conserved NPxxY motif (N^7.49^P^7.50^xxY^7.53^; superscripts are Ballesteros–Weinstein numbering for GPCRs, where the first number before the dot indicates the number of the TM in which the residue is located, and the second number shows the position of the residue relative to one of the most conserved residues in that TM, which is arbitrarily assigned the number 50 [75]). While α-carbon root-mean-square deviation (RMSD) values between matched residues in other conserved motifs, D^3.49^R^3.50^Y^3.51^ and P^5.50^I/V^3.40^F^6.44^, are less than 2.5 Å, R121 of the E/DRY motif adopts a different rotamer, with a distance of about 7.8 Å between the NH1 atoms of R121 in the two structures. A similar conformational rearrangement is detected between the inactive M_2_R structure and the agonist-bound active state of the M_2_R stabilized by a G protein mimetic camelid antibody fragment (nanobody) [76]. Furthermore, D120 in the DRY motif is stabilized via a hydrogen bond with N58^2.39^ in the active state [76]. The movement of Y440^7.53^ in the NPxxY motif brings it into close proximity to the highly conserved Y206^5.58^, allowing water-mediated hydrogen bond formation similar to that of active β_2_AR [77] and rhodopsin [78]. All existing data suggest that the opening of the cavity on the cytoplasmic side is a hallmark of GPCR activation necessary for receptor coupling to G proteins and other signal transducers [76]. It has been shown that the triple mutation T68F^2.39^, Y132G^3.51^, and Y219A^5.58^ in TMs 2, 3, and 5 in β_2_AR abrogated G protein coupling while maintaining arrestin-mediated signaling [79]. Furthermore, S^5.42^, S^5.43^, and S^5.46^ in the TM5 of β_1_AR [80], β_2_AR [77], and dopamine D_2L_ receptors [81] are critical for G protein coupling, and disruption of ligand contacts with these residues precludes G protein coupling while preserving arrestin-mediated signaling. Existing data suggest that the size of the interhelical cavity and several residues in the receptor and G proteins that engage the partner determine the selectivity of the interaction of that GPCR with a particular subfamily of G proteins [82,83,84]. While bound arrestins in the complex engage different residues in the interhelical cavity than the cognate G proteins, it appears that bound arrestins fit best when the size of the cavity is intermediate between the larger G_s_-binding and the smaller G_i_-binding cavity. Thus, the size of this cavity per se is unlikely to determine receptor selectivity for arrestins vs. G proteins. However, the structures of too few GPCRs in complex with both G proteins and arrestins have been solved to enable an unambiguous conclusion.

Arrestins appear to engage a smaller area on the cytoplasmic receptor surface than G proteins. The total area buried in the rhodopsin-arrestin-1 interface (1350 square Å) is much smaller than the interface area of the β_2_AR-G_s_ complex (2576 square Å) [40], but in the case of arrestin, it is augmented by the binding of the phosphorylated receptor C-terminus to the N-domain of arrestin [43]. The interface area of the phosphorylated β_1_AR-arrestin-2 complex (~1200 square Å, excluding the interface of the C-terminal phosphopeptide of the vasopressin V_2_ receptor, V_2_Rpp) is even smaller than that of rhodopsin-arrestin-1 [49]. Unlike the contiguous interface observed in the β_2_AR-G_s_ complex, the rhodopsin-arrestin-1 complex has four distinct interface patches. A structural rearrangement upon activation allows the finger loop of arrestin-1 (Q70 to L78) to form an extensive interaction with the C-terminus of TM7, the N-terminus of H8, and the elements of ICL1 of rhodopsin, thereby forming the first patch. Second, the N-domain β-strand VI (residues 79–89) that follows the finger loop interacts with residues from TM5, TM6, and ICL3. Third, residues in the arrestin-1 back loop (R319 and T320) interact with the C-terminus of TM5. The fourth interface patch is mainly between the arrestin-1 N-terminal β-strand I (residues 11–19) and the C-terminus of rhodopsin [40]. Extensive mutations in ICL1 (L68R, T270R, P71W, L72R, and L76R), TM3 (E134W, R135E, R135G, R135W, V138R, and V139R), ICL2 (K141R, N145G, and F146G), TM5 (V230R and A233R), TM6 (V250R), and TM7 (Y306G and N310R) and deletion of H8 were shown to significantly reduce arrestin-1 binding to rhodopsin [40]. It is conceivable that the interrupted (patchy) arrestin–receptor interface helps arrestins to accommodate active GPCRs with different sizes of the interhelical cavity, whereas the contiguous G protein–receptor interface makes G proteins of different subfamilies more selective for a particular size of the cavity in the active receptor. This idea must be tested by obtaining the structures of G protein and arrestin complexes of GPCRs that couple to the other two G protein subfamilies, G_q/11_ and G_12/13_. None of these structures are available yet.

The next important question is, do G proteins and arrestins bind the same active conformation of a GPCR? Comparison of the structures of active M_2_R bound to G_oA_ (PDB: 6OIK) [74] with M_2_R bound to arrestin-2 (PDB: 6U1N) [33] reveals similar active conformations. However, relatively smaller shifts in the cytoplasmic ends of TM7 and helix 8, but larger shifts in the cytoplasmic ends of TM5 (~2.7 Å for K212) and TM6 (~3.2 Å for K383), as well as the extracellular ends of TM5, ECL2, and ECL3 were revealed in M_2_R bound to arrestin-2, as compared to its complex with G_oA_ (Figure 4G–I). The movement of TM6 in G_s_-coupled β_1_AR containing the phosphorylated C-terminus of vasopressin receptor 2 in complex with arrestin-2 is about 7 Å smaller than that in the β_2_AR-G_s_ complex [49]. Thus, the intracellular cavity needed for a GPCR to accommodate arrestin-2 is slightly larger than that needed for G_αi_ and slightly smaller than that required for G_αs_. Assuming that GPCRs interact with G proteins prior to interaction with arrestins, an interhelical cavity as narrow as that required for G_αi_ seems to be sufficient for arrestin to enter this cavity and subsequently induce a larger displacement of TM6. This, however, is a preliminary conclusion which is drawn based on the comparison of M_2_R and β_1_AR only. Additional structures of other GPCR complexes with different transducers will certainly clarify the mechanisms that determine transducer selectivity. For instance, the promiscuous NTS_1_R (which couples with G_s_, G_q/11_, G_i/o_, and G_12/13_) in complex with truncated arrestin-2 shows a similar displacement of TM6 when compared to its canonical complex with G_i_ but a slightly larger outward movement of TM5 (~2.1 Å for Cα of A270) [34].

The movement of TM6 in GPCR-G_i/o_ complexes is also smaller than that in the β_2_AR-G_s_ complex [65,72,85]. Given that the reverse turn in the C-terminus of G_αs_ is bulkier than that of G_αi_, cation–π interactions between R^3.50^ in TM3 of the G_s_-coupled receptors and the α5 helix of G_αs_ requires about 6 Å larger outward movement of TM6 compared to the hydrogen bond between the corresponding elements in G_i_-coupled receptors interacting with G_αi_ [86]. A similar phenomenon in the arrestin finger loop interaction with receptor elements can explain the different extent of TM6 movement in the receptor–arrestin complexes. While the majority of amino acid side chains at the phosphorylated β_1_AR-arrestin-2 and β_2_AR-G_s_ interfaces are similarly positioned, R^3.50^ (TM3) in the E/DRY motif adopts different rotamers, resulting in different contact formation. In fact, R^3.50^ in β_2_AR-G_s_ [64] and adenosine A_2A_ receptor-G_s_ complexes [87] extends away from TM3 and forms contact with the hydrophobic cavity of the receptor and Y391 in the α5 helix of the G protein. In contrast, R^3.50^ in the β_1_AR–arrestin-2 complex adopts a different rotamer, allowing extensive polar contacts with D138^3.51^ and T76^2.39^ in the receptor and with D69 of the finger loop of arrestin-2 [49]. Thus, arrestin-2 appears to bind GPCRs in conformations that are similar but not identical to those that engage the cognate G proteins. This is consistent with the idea that different ligands can bias a GPCR towards G protein or arrestin. However, the relatively minor differences in the conformations of the same receptor bound to the cognate G protein and arrestin-1 indicate that the achievable degree of such bias is limited (reviewed in [88]).

Structural data of GPCRs bound to biased ligands in the presence of cytosolic partners are scarce. To our knowledge, only one study has provided a comparison between formoterol (which displays bias toward arrestin over G_s_ compared to isoproterenol)-bound β_1_AR-arrestin-2 complex and formoterol-bound β_1_AR-nanobody 80 (Nb80, which was used as a G_s_ mimetic) [49]. The most notable differences between these structures were a closer position of the cytoplasmic ends of TM5 (6.7 Å for Cα of I241) and TM6 (1.9 Å for Cα of K284) to the receptor core in the former structure as well as a 2.2-Å shift of the ECL2 position upon arrestin-2 binding. Furthermore, formoterol seems to lose a substantial fraction of its contacts with TM5 and TM3 as compared to unbiased agonists, forming additional interactions with TM6 [49]. Crystal structures of β_1_AR [89] and 5-HT_2B_ receptors [90] bound to biased ligands in the absence of cytoplasmic partners have also suggested a role for specific contacts of the ligand with ECL2, TM5, TM6, and TM7 as potential requisites for arrestin-biased activity. The amino acid difference in the 7.35 position between μ- and κ-opioid receptors, W^7.35^ and Y^7.35^, respectively, might be the reason why the same ligand can facilitate arrestin-2 recruitment only to the latter receptor [91]. Interaction with this residue was also found to be critical for M_2_R [92] and β_2_AR [93] ligands in terms of signal direction toward different transducers. Although crystal structures of the angiotensin AT_1_R bound to unbiased and arrestin-biased ligands demonstrate similar binding poses at the extracellular face, they show remarkable differences at the base of the ligand-binding pocket, as well as at the receptor core. In particular, arrestin-biased ligands induce a shift of TM6 and a conformational rearrangement in N295^7.46^, whereas the unbiased angiotensin II causes an additional outward movement of N111^3.35^ in TM3, suggesting conformational changes of N295^7.46^ in TM7 as the trigger for the additional rearrangement needed for arrestin-2 recruitment and movement of N111^3.35^ as the prerequisite for G protein coupling [94]. It is noteworthy that the majority of GPCRs with a sodium-binding site, which stabilizes the positions of TM3 and TM7 relative to each other in the inactive conformation and collapses upon activation, contain highly conserved serine in the 7.46 position, whereas AT_1_R, which does not interact with sodium, contains asparagine (with a bulkier side chain than serine) in this position. In fact, the relative positions of TM3 and TM7 in the inactive AT_1_R are stabilized via a hydrogen bond between N295^7.46^ and N111^3.35^, which is disrupted upon angiotensin II binding [94]. Interestingly, mutations in the sodium-binding site or surrounding residues in δ-opioid [95], chemokine CCR_5_ [96], and neurokinin NK_1_ [97] receptors induce biased signaling toward either G proteins or arrestins. However, we do not have enough structural information to predict the direction of the bias induced by a particular mutation.

Based on existing structures, as well as biophysical [22,39] and biochemical [27,98,99,100] evidence, the arrestin finger loop (purple in Figure 1B) directly interacts with the interhelical cavity of active GPCRs, similar to the α5 helix of the Ras domain in G_s_ in GPCR-G protein complexes, albeit at a slightly lower depth and via a relatively smaller interaction interface. Thus, the finger loop likely serves as the arrestin sensor of receptor activation [100]. In fact, the finger loop, especially Leu71 in arrestin-2, provides the majority (37%) of total contacts with the β_1_AR, mainly with TM6, TM2, and ICL2 [49]. While the interhelical cavity shows a conserved hydrophobic core, it is surrounded by charged residues that are less conserved. In different GPCRs, it must have differentially spaced charges that are also engaged by the finger loop: the replacement of the same leucine with residues with positively and negatively charged side chains differentially affected arrestin-3 binding to M_2_ muscarinic and D_2_ dopamine receptors [27]; the elimination of two positive and three negative charges in the finger loop of arrestin-1 significantly reduced its binding to light-activated unphosphorylated rhodopsin [100]. In agreement with the participation of the finger loop charges in binding, in the complex of arrestin-2 with M_2_R, D69 in the arrestin-2 finger loop forms an H-bond with receptor residue N58 and a possible salt bridge with another receptor residue, R121 [33]. Importantly, D69A mutation reduced arrestin-2 binding to M_2_R [33], supporting its role in this interaction. The finger loop, C-loop, lariat loop, and middle loop are parts of the arrestin side of the interface in the NTS_1_R-arrestin-2, M_2_R-arrestin-2, and rhodopsin-arrestin-1 complexes; however, these arrestin elements interact with different parts of the receptors (Figure 3). Notably, the finger loop in the NTS_1_R-arrestin-2 complex extends further away from the N-domain of arrestin-2 compared to that of arrestin-1 in the rhodopsin-arrestin-1 complex or that of arrestin-2 in the M_2_R-arrestin-2 complex (Figure 2) [34], highlighting the flexibility of the finger loop that enables arrestin-2 to adopt different orientations in complex with different receptors (Figure 3).

To summarize, the cavity between the transmembrane helices that opens upon GPCR activation is a common clue used by all proteins that preferentially bind active receptors. All existing biochemical, biophysical, and structural data show that arrestins interact with this cavity via the finger loop in the central crest of their receptor-binding side (Figure 1). Apparently, arrestins are able to efficiently bind active GPCRs with a larger (as in G_s_-coupled receptors) or smaller (as in G_i_-coupled receptors) cavity. The role of receptor charges in and around this interhelical cavity in arrestin binding needs to be investigated more extensively. Reduction in the flexibility of the finger loop was shown to dramatically reduce arrestin-3 binding to all GPCRs tested [27]. Sequence conservation in GPCRs is limited [14], yet the two non-visual arrestins in vertebrates apparently interact with hundreds of distinct GPCR subtypes [5]. Different receptors likely have distinct spacing of the charges in the interhelical cavity, which can force the turning of the highly flexible arrestin finger loop in different ways in the complex. It is still not clear whether this difference leaves a distinct imprint on the conformation of the opposite side of the arrestin molecule, where the effectors bind (Figure 1) [12], i.e., whether the identity of the receptor that arrestin interacts with is encoded in the conformation of bound arrestin.

## 6. Receptor Intracellular Loops: What Role Do They Play in Arrestin Binding?

Each GPCR has three intracellular loops (ICLs), which are numbered ICL1–3, starting with the most upstream one between TM1 and TM2. These loops face the cytoplasm and their conformation in active and inactive receptors is different. Thus, it is only natural that all intracellular transducers of GPCR signals, G proteins, GRKs, and arrestins, engage residues in these loops. The importance of several residues in ICL1 and ICL2 of rhodopsin for arrestin-1 binding was demonstrated by mutagenesis [101,102]. It was shown that T70^ICL1^C and K67^ICL1^C mutations in rhodopsin allow disulfide bond formation with Q70C, E71C, and D72C (in the N-terminus of the finger loop) in arrestin-1 [40]. Thus, these residues are likely located in close proximity in the complex of native proteins, allowing an interaction between them. In agreement with these observations, native Cys residue 217 in ICL1 of the parathyroid hormone receptor (PTH_1_R) strongly cross-linked in cells upon receptor activation with the unnatural amino acid O-(2-bromoethyl)-tyrosine (BrEtY) in position 66 in the N-terminal part of the finger loop of arrestin-2 [103]. Thus, ICL1 is another receptor element, in addition to the interhelical cavity, that engages the finger loop of arrestin proteins.

Direct interaction of ICL2 with receptor-bound arrestins was revealed in the structures of arrestin-1 complex with rhodopsin [40] as well as arrestin-2 complex with NTS_1_R [34], M_2_R [33], and β_1_AR [49]. In fact, ICL2 of β_1_AR, particularly F147^34.51^, Q150^34.54^, T154^4.38^, and R155^4.39^, provides the majority (45%) of total contacts with arrestin-2, mainly with β-strand V, the finger loop, β-strand XV, and the loop between β-strands XVII and XVIII [49]. Furthermore, the G149C mutation in rhodopsin ICL2 allows disulfide cross-linking with the D139C mutant of arrestin-1, indicating that G149^ICL2^ is located in close proximity to arrestin-1 D139 (mouse numbering; D138 in bovine arrestin-1; these residues are in the loop between the short β-strands VIII and IX in arrestin-1 [16]), allowing an interaction between them [40]. Notably, this loop (termed 139 loop in arrestin-1 [104] and middle loop in arrestin-2 [47]) was shown by intra-arrestin distance measurements to shift more than any other arrestin element upon receptor binding [37,104] and is invariably found in contact with the receptor in the structures of arrestin–receptor complexes [32,33,34,40,43,49]. While early studies showed that E/DRY^ICL2^/AAY mutants of angiotensin AT_1A_R do not couple to G protein but still signal via arrestins [105], detailed examination of these mutants, as well as histamine H_1_ receptor mutated in the same region, using a panel of dynamic live cell biosensor assays revealed efficient G protein signaling [106]. It is noteworthy that ICL2 is involved in receptor interaction with both arrestins and G proteins. In the latter case, key interactions are demonstrated between residues in the receptor ICL2 and the αN or α5 helices of G_α_, and a role for specific residues in this loop is suggested for the selectivity of coupling to G_i_ vs. G_s_ (reviewed in [83]). In particular, the conserved hydrophobic residue within the ICL2 (position 34.51) may not be involved in the primary coupling of M_2_R to G_i/o_ proteins. However, this region is critical for the secondary coupling of β_2_AR to G_i/o_ proteins after phosphorylation by protein kinase A (PKA), although unphosphorylated β_2_AR selectively engages G_s_ [107]. The position of ICL2 with respect to the interacting elements of arrestins varies in different structures. In the NTS_1_R-arrestin-2 complex [34], ICL2 is located on the outer side of the C-loop, while in the rhodopsin-arrestin-1 [43] and M_2_R-arrestin-2 [33] complexes, it is sandwiched in the cleft between the N- and C-domains that is comprised of portions of the finger, middle, gate, and C-loops (Figure 3). This orientation of ICL2 is mainly determined by hydrophobic interactions between Leu129 (and possibly other hydrophobic residues in other GPCRs) with the arrestin-2 hydrophobic cleft [33,34]. The movement of the middle and lariat loops upon recruitment to the receptor creates a cleft that accommodates ICL2 of rhodopsin, allowing interaction with the middle (Val140 region) and lariat (Tyr251 region) loops [40]. Thus, ICL2 of GPCRs participates in their interactions with both G proteins and arrestins.

Part of rhodopsin ICL3 was also found in contact with the back loop of arrestin-1 [40]. ICL3 of NTS_1_R also interacted with arrestin-2 [34], even though arrestin-2 was turned almost 90^o^ relative to the receptor, as compared to the rhodopsin-arrestin-1 or M_2_R-arrestin-2 complexes (Figure 3). Native Cys residue 397 in the ICL3 of PTHR1 strongly cross-linked upon receptor activation in cells with BrEtY incorporated in arrestin-2 position 78 in the C-terminal part of the finger loop [103]. The positioning of the N-terminal part of the finger loop near ICL1 and of the C-terminal part near ICL3 of PTHR1 is consistent with the orientation of arrestin-2 relative to the receptor in the complex with M_2_R and arrestin-1 in complex with rhodopsin, but not with the orientation of arrestin-2 bound to NTS1 [103], suggesting that arrestins bind PTHR1 in the same orientation as rhodopsin and muscarinic M_2_ receptor. M_2_R in complex with arrestin-2 was fused with V_2_Rpp. This added C-terminus interacted with the phosphate-binding arrestin-2 elements essentially as the separated V_2_Rpp in complex with truncated arrestin-2 [47]. Thus, that structure did not reveal where native phosphorylation sites located on the ICL3 of M_2_R [108,109] bind. An electron density map revealed strong density for the C-terminus of ICL3 in the NTS_1_R-arrestin-2 complex. Notably, S287^ICL3^, which is phosphorylated in this structure, is located adjacent to R76 and K77 of the arrestin-2 finger loop. This suggests that phosphorylation of this residue, or potentially any other phosphorylatable residue in ICL3, may be involved in the disruption of K77 contact with E313, facilitating the shift from the inactive to the active arrestin conformation [34]. Furthermore, the Q237C mutation in rhodopsin ICL3 permits disulfide cross-linking with R319C and T320C (back loop) arrestin-1 mutants, indicating sufficiently close proximity of rhodopsin ICL3 to the arrestin-1 back loop in the complex to permit an interaction between them [40]. It is noteworthy that ICL3 of μ-opioid receptor [65] and β_2_AR [64] makes hydrophobic interactions with G_i_ and G_s_, respectively. Polar interactions between ICL3 of μ-opioid receptor and the β6 sheet of G_i_ are absent in the β_2_AR-G_s_ complex. These data, along with mutational studies, suggest a possible role for specific residues in this loop for selective coupling to particular G proteins (reviewed in [83]). Thus, ICL3 participates in arrestin binding and likely differentially engages G proteins of different subfamilies.

While there is a consensus in the field that the ICLs of GPCRs play a role not only in arrestin binding but also in G protein coupling and GRK activation, experimental data are scarce. We need more structures of arrestin, GRK, and G protein complexes with different GPCRs, along with complementary receptor mutagenesis studies, to obtain a more comprehensive picture of specific contacts between these interacting molecules, which will enable us to elucidate the potential of manipulating these elements for therapeutic purposes.

## 7. The Role of Helix 8 in Arrestin Binding

In many GPCRs, the proximal part of the C-terminus between TM7 and the palmitoylation site forms the short helix 8 parallel to the plane of the membrane (Figure 4). Thus, helix 8, along with the ICLs, is part of the cytoplasmic “face” of GPCRs that is recognized by intracellular signal transducers. Not surprisingly, helix 8 was shown to play a role in arrestin-1 binding [110]. Interestingly, biophysical studies using fluorescently labeled purified vasopressin V_2_ receptor [111] and β_2_AR labeled with ^19^F-containing NMR probes [112] found that G protein-biased agonists perturb TM6 and arrestin-biased agonists perturb TM7 and helix 8, whereas unbiased agonists induce both perturbations. Furthermore, dopamine D_1_ receptor mutated in helix 8 displays enhanced G protein signaling but reduced arrestin-mediated desensitization [113]. These data suggest that helix 8 and the TM7 immediately preceding it play an important role in arrestin binding. Indeed, helix 8 of rhodopsin was found to contact the finger loop of arrestin-1 in the complex [40]. The fact that N310^7.57^C and Q312^8.49^C mutations allow disulfide bond formation with G77C (in the mouse arrestin-1 finger loop, homologous to G76 in bovine arrestin-1) supports this notion [40]. However, comparison of the M_2_R-G_oA_ (PDB: 6OIK) [74] and M_2_R-arrestin-2 (PDB: 6U1N) [33] complexes reveals relatively small differences in the positions of the cytoplasmic ends of TM7 and helix 8 (Figure 4G–I). Moreover, in complexes with NTS_1_R [32,34] and M_2_R [33], arrestin-2 did not contact helix 8 of these receptors. In particular, the electron density map excludes S396, S401, S403, and S404 of NTS_1_R from being in contact with K294 (in the gate loop) of truncated arrestin-2, leaving only phosphorylated T407 as a candidate residue to form this bond that would explain the density observed in the NTS_1_R C-terminus and arrestin-2 N-domain [34]. Similarly, T491 and T360 on the C-terminus of M_2_R and V_2_Rpp, respectively, establish interactions with R25 (N-domain) and K294 (gate loop) of arrestins, aiding in the interdomain twisting [33]. In the case of PTHR1, native cysteines 460 and 462 in the short connector between TM7 and H8 did not significantly cross-link with BrEtY residues in various positions in the arrestin-2 or arrestin-3 finger loop [103]. Thus, helix 8 of some receptors engages bound arrestin, but this does not appear to be a universal rule.

## 8. Receptor-Attached Phosphates in Arrestin Binding

As a rule, arrestins bind active phosphorylated receptors much better than all other functional forms: active non-phosphorylated, inactive phosphorylated, or inactive non-phosphorylated [20,23,24,58,59,114,115,116]. Thus, there must be a molecular mechanism whereby receptor-attached phosphates “activate” arrestins, triggering high-affinity binding; e.g., arrestins must have a “phosphate sensor”—an element that binds receptor-attached phosphate(s) in such a way that phosphate binding converts the sensor from an inhibitor of arrestin action in the absence of the bound phosphate to an activator in its presence.

The first candidate for the role of the phosphate sensor in arrestins was discovered before any structures were solved: R175 in bovine arrestin-1 was shown to bind rhodopsin-attached phosphates [117]. Comprehensive mutagenesis of this residue showed that charge neutralization or reversal invariably enhanced arrestin-1 binding to non-phosphorylated light-activated rhodopsin (Rh*), as if the sensor was artificially turned “on” by these mutations, whereas its conservative substitution with a positively charged lysine preserved high arrestin-1 selectivity for phosphorylated light-activated rhodopsin (P-Rh*) [117,118]. The most parsimonious explanation of the data was a hypothesis that in the protein, this arginine interacts with a negatively charged partner, and that phosphate binding (or mutation) disrupts this interaction, “informing” arrestin-1 that the phosphate is in place (reviewed in [2]). Crystal structures of basal arrestin-1 revealed that this arginine is an integral part of the “polar core”, an unusual (for a soluble protein) arrangement of five interacting charged residues in the middle of the molecule that are virtually solvent-excluded [16,119]. Homologous arginines in the structures of other arrestin subtypes were found in similar polar cores [17,18,20,35], demonstrating that this feature is conserved in the family. The five polar core charges are remarkably conserved in arrestin evolution [5,15]. Naturally, the three aspartic acids present in the polar core emerged as prime candidates for the role of the negatively charged partners that this arginine interacts with in the basal state. Individual replacement of these aspartates with positively charged arginines identified D296 as the key partner: the charge reversal D296R mutation in bovine arrestin-1 resulted in virtually the same phosphorylation-independent binding as R175E [16,28]. Most importantly, simultaneous charge reversal of both residues, restoring the salt bridge, also restored wild-type (WT) arrestin-1 selectivity for P-Rh* [16,28]. Charge reversal of homologous arginines in non-visual arrestin-2 and -3 also greatly enhanced the binding to unphosphorylated forms of their cognate receptors [23,24,29,30,31], as could be expected if this phosphate-sensing mechanism is shared by all members of the arrestin family. Thus, the issue appeared to be settled: R175 and its homologues in other subtypes act as phosphate sensors—phosphate binding neutralizes the charge of the arginine, breaking the salt bridge with the homologue of D296, which destabilizes the polar core, thereby “telling” arrestin to swing into action (reviewed in [2]). Importantly, this mechanism was independent of the sequence context of the phosphorylated receptor residue, which satisfactorily explained how very few arrestins can recognize the phosphorylation state of numerous GPCRs with very low conservation of the sequence of intracellular elements.

However, solved structures of arrestin complexes with phosphorylated receptors buried this intellectually pleasing model. Receptor-attached phosphates did not interact with the polar core arginine in arrestin-1 bound to rhodopsin [43], or in arrestin-2 bound to NTS_1_R [32,34], M_2_R [33], and β1-adrenergic receptor [49]. These structures, as well as the structure of the arrestin-3 trimer in complex with inositol-hexaphosphate (IP_6_) [27] where each protomer was in the “active” receptor-bound-like conformation, suggested another candidate. A lysine located between two polar core aspartates on the gate loop (K294 in arrestin-2, dark red in Figure 1B) was invariably found in contact with one of the phosphates of the activator. Thus, it appeared reasonable to hypothesize that phosphate binding to this conserved lysine pulls the gate loop out of place, destabilizing the polar core by the removal of the two negative charges [11]. This model did not survive experimental testing either. Elimination or even reversal of the charge of this lysine in bovine and mouse arrestin-1, as well as in arrestin-2 and -3, did not dramatically reduce their binding to the cognate receptors, as this model predicted [31].

At least one of the pairs of lysines in β-strand I (K11 or K12 in arrestin-2, red in Figure 1B), also conserved in all arrestins [5,15], interacts with the phosphate attached to the phosphorylated receptor or IP_6_ in the structures [27,32,33,34,43]. In all arrestins in their basal state, this β-strand is part of the three-element interaction involving α-helix I of the N-domain and β-strand XX of the C-tail, which anchors the C-tail (light blue in Figure 1B) to the N-domain. Phosphate binding to either of these lysines would shift β-strand I out of its basal position, destabilizing the three-element interaction, which would result in the release of the C-tail, which supplies another arginine to the polar core. Receptor binding and arrestin activation were shown to involve the release of the arrestin C-tail using a variety of methods [37,38,39,120,121,122]. Indeed, forced release of the C-tail by triple alanine substitution of conserved bulky hydrophobic residues that interact with α-helix I and β-strand I, as well as C-tail deletion, were shown to enhance the binding of all arrestins to unphosphorylated receptors [23,24,29,30,123]. Consistent with the hypothesis that one or both lysines in β-strand I serve as a phosphate sensor, their replacement with alanines reduced in vitro binding of WT arrestin-1 to purified P-Rh* by ~90%, but had little effect on the binding of “pre-activated” mutants, where the polar core or three-element interaction were independently destabilized by other structurally distinct mutations [36]. The same double alanine substitution greatly reduced in-cell arrestin-1 binding to non-cognate M_2_R and β_2_AR [25], where phosphate interaction was likely the main driving force. Thus, in arrestin-1, one or both lysines in β-strand I perform exactly as the phosphate sensor is expected to: the elimination of the charge essentially obliterates the binding of WT protein to P-Rh* but does not affect the binding of enhanced forms where the polar core or three-element interaction were destroyed by other mutations [36]. However, alanine substitution of the two homologous lysines in arrestin-2 and -3 did not result in a dramatic reduction in their in-cell binding to M_2_R or β_2_AR [25], or in the binding of arrestin-3 to the neuropeptide Y receptors Y_1_ and Y_2_ [56]. Thus, the phosphate sensor in non-visual isoforms, which are a lot less selective for phosphorylated receptors than arrestin-1 [23,24,30,115], does not appear to play as critical a role in receptor binding as in the exceptionally phosphorylation-sensitive arrestin-1.

The next important issue is how many phosphates must be attached to a GPCR to make it a preferred target for arrestins. A study using genetically modified mice suggested that rhodopsin must have at least three phosphorylation sites for rapid engagement of arrestin-1 that quenches its light-induced signaling [124]. An in vitro study of purified rhodopsin species with different levels of phosphorylation, which were confirmed by mass spectrometry, also indicated that three phosphates are necessary for high affinity arrestin-1 binding [125]. Human and mouse rhodopsin has six potentially phosphorylatable serines and threonines in its C-terminus, but it appears that from the arrestin-1 standpoint, not all sites are equal [126,127]. High-resolution structure of the arrestin-1 complex with rhodopsin revealed that arrestin-1 has three positively charged pockets suitable for the binding of phosphates or receptor residues with negatively charged side chains. The three positively charged pockets are formed by basic residues: K16, R19, and R172 (pocket A), K16, R30, and K301 (pocket B), and K15 and K111 (pocket C), which, in the structure, accommodate phosphates attached to T336 and S338 and the negatively charged E341, respectively, of the rhodopsin C-terminus [43]. The majority of GPCRs have potential phosphorylation sites adequately spaced to fit these pockets, which are conserved in other arrestin subtypes [43]. Interestingly, β_2_AR, which binds non-visual arrestins transiently [128], does not have a complete “phosphorylation code”, and its creation by receptor mutagenesis greatly enhanced arrestin binding to this receptor [43,129]. Importantly, the structure suggests that one of these arrestin pockets can be occupied by a negatively charged receptor residue [43], which is consistent with earlier findings that some negative charges in GPCRs are critical for arrestin binding [130]. The crystal structure of the CXCR7 phosphopeptide (C7pp) complex with arrestin-3 reveals a similar interaction between the second and third phosphorylated residues (pT338 and pT341) with pockets A and B, but a new pocket (pocket P around R148) was found to engage the first phosphorylated residue (pS335). This newly found pocket can accommodate one (PxPxxP) or two (PxxPxxP) residues between the first and second phosphorylated residues and may partly explain the difference between the C7pp-arrestin-3 complex and other complexes—e.g., a smaller interdomain rotation and different conformations of loops. Furthermore, the positively charged residues in arrestin-2 (R8, K10, K11, K107, K294) that interact with phosphorylated residues in V_2_Rpp are different from those involved in the formation of the C7pp-arrestin-3 complex [44]. Thus, while overwhelming evidence suggests that more than one receptor-attached phosphate is required for high-affinity arrestin binding in most cases, there does not appear to be a generalizable rule on the number or spacing of these phosphates, nor on how many of them can be substituted by negatively charged receptor residues. Moreover, the relative role of receptor-attached phosphates in arrestin binding differs among GPCRs, with a strict requirement in certain receptors (e.g., rhodopsin, β2AR), whereas some other receptors can bind arrestins in the absence of phosphorylation.

The phosphates that promote arrestin binding are, in most cases, attached by GRKs, although there are reports that phosphates attached by other kinases can also “attract” arrestins [131,132]. Usually, GRKs selectively phosphorylate active GPCRs, likely because GRKs are activated by binding to active receptors [133,134,135]. The idea that GRK activation appears to require its physical interaction with an active receptor is supported by structural studies [70,71]. Thus, the reported activity of members of the GRK4 subfamily towards inactive GPCRs [136] is likely explained by their ability to dock to the inactive receptors and induce their transition into an active-like conformation capable of activating these GRKs. The well-established inherent flexibility of GPCRs [137,138] suggests that the initial interaction does not even need to be strong to shift the conformational equilibrium of the receptor.

A great majority of GPCRs have a lot more than the three required phosphorylatable serines and threonines in their intracellular elements and often contain more than one potential phosphorylation code [43]. The phosphates to which arrestins bind can be localized on the receptor C-terminus, as in rhodopsin [124], β_2_AR [139,140,141,142], Y_2_ neuropeptide Y receptor [143], and many other GPCRs. However, in some receptors, the phosphates necessary for arrestin binding are localized in the third [108,109] or second [144] cytoplasmic loop. The necessity of phosphorylated residues at transmembrane regions, such as Y219^5.58^ of β_2_AR, for arrestin recruitment to the receptor suggests a role in arrestin binding, either direct or indirect, for phosphorylated residues buried deeply in the interhelical cavity [52].

The sheer number of serines and threonines on the cytoplasmic elements of most GPCRs that can be phosphorylated and “attract” arrestins led to the idea that the pattern of phosphorylation determines its biological consequences [53], which later came to be known as the “barcode” hypothesis [145]. The original hypothesis explicitly suggested that the positions of the phosphorylated sites on the same receptor in different cell types and/or under different physiological conditions in the same cell type determines the conformation of bound arrestin and, therefore, the consequences of arrestin binding to these differentially phosphorylated receptors [53]. Indeed, the phosphorylation of the same receptor by different GRKs, presumably at different positions, was shown to elicit distinct biological responses [50,51,52]. This model implies that arrestins bound to the same differentially phosphorylated receptor can assume distinct conformations. This idea was proposed more than a decade ago [5] and appears to be supported by the remarkable differences in the “poses” of arrestins bound to different receptors in the recently solved structures of the complex [32,33,34,40]. The analysis of the structural perturbations in arrestin-1 induced by differentially phosphorylated rhodopsin C-terminal peptides [127] also supports this notion. However, we should keep in mind that the binding of the phosphopeptide is not the same as the binding of the full-length phosphorylated receptor. The ability of arrestins to assume different conformations when bound to the same differentially phosphorylated receptor still needs to be demonstrated experimentally. In this case, another level of complexity will be added to the capacity of GPCR signaling. In particular, a given GPCR may signal through several transducers, including multiple G proteins and two non-visual arrestins, and the capacity of a biased ligand to direct a signal one way over another is a hot topic in the discovery of GPCR-targeting drugs [83,146,147]. If different conformations of arrestin when bound to differentially phosphorylated receptors result in distinct biological outcomes, there is one more way to create signaling selectivity at this junction, suggesting how pharmacological intervention can direct a signal in favor of or away from particular pathways.

## 9. Missing Pieces

Experimental data identifying the receptor residues participating in arrestin binding are far from comprehensive in the case of rhodopsin [101,102], fragmental in the case of dopamine D_1_ receptor [54], and virtually absent for hundreds of other GPCRs. Moreover, current structures of GPCRs occupied by a biased ligand are limited to only one receptor in the presence of cytoplasmic signal transducers and to a handful of receptors in the absence of cytoplasmic partners. To be comprehensive, this work needs to be done with numerous receptors, as GPCRs demonstrate relatively low homology in their intracellular elements [14], which suggests that if a certain residue in one receptor binds arrestins, that does not mean that the corresponding element in another GPCR also participates in this interaction. For instance, phospho-T359 does not make any contacts with arrestin-2 in the V_2_Rpp arrestin-2 complex, but it interacts with the lariat loop (K294 and H295) of arrestin-2 in the phosphorylated β_1_AR-arrestin-2 complex. The N-terminal region of V_2_Rpp is located in close proximity to the finger loop of arrestin-2, while no density is observed in the corresponding region in the β_1_AR-arrestin-2 complex [49].

Most vertebrates have two non-visual arrestins (bony fish that underwent an extra whole-genome duplication have three [15]), whereas invertebrates apparently express only one non-visual isoform. As the duality of the non-visual arrestin subtypes has been preserved for at least 400 million years in vertebrate evolution from fish to mammals, the two subtypes must have different functional capabilities. For instance, while arrestin-2 is involved in the desensitization of δ-opioid receptors and the development of tolerance, arrestin-3 acts as a facilitator of resensitization and an inhibitor of tolerance [148]. While the majority of known non-GPCR binding partners interact with both arrestin-2 and -3, there are proteins that bind one subtype but not the other [4]. As far as signaling is concerned, the best-established qualitative difference between non-visual arrestins is the ability of arrestin-3, but not the highly homologous arrestin-2, to facilitate the activation of c-Jun N-terminal kinases (JNKs) [19,149,150,151,152]. In terms of GPCR binding, the two non-visual subtypes are also somewhat different. Many GPCRs prefer arrestin-3 over arrestin-2 [128], although there are receptors with the opposite preference [153,154]. So far, no clue has emerged as to why some receptors prefer one non-visual arrestin over the other, and no attempt at changing the arrestin preference of a particular GPCR has been made. 

Even though the subject of this review is the mechanisms of the interactions between arrestins and GPCRs, arrestin-mediated signaling deserves a note. This subject became unduly controversial, largely because of the focus on just one branch of signaling out of many. The claim that there is G protein-independent arrestin-mediated activation of extracellular signal-regulated kinases (ERK)1/2 [79] was recently questioned based on the findings that ERK activation requires functional G proteins [155,156] and that ERK1/2 can be activated by β2AR in the absence of both non-visual arrestins [157]. However, even in the case of ERK1/2, an arrestin contribution was reported [156,158], suggesting that arrestins might not initiate signaling in the Raf1-MEK1-ERK1/2 cascade, but play a role in its propagation (reviewed in [159]). It is important to remember that arrestins bind numerous signaling molecules [4], not only members of the Raf1-MEK1-ERK1/2 cascade. Arrestins have been reported to facilitate the activation of c-Src [160], p38 [161], and JNK3 [149], among others. The role of G proteins in the activation of these branches of signaling has never been experimentally tested. Moreover, numerous lines of evidence suggest that GPCRs do not play a role in arrestin-3-dependent JNK3 activation (e.g., [19,162]), which suggests that GPCR-activated G proteins are unlikely to be involved. Thus, the relative role of arrestins and G proteins must be investigated in each branch of signaling separately, as it is highly unlikely that the same rules apply in every one of them.

Another issue which might be even more important conceptually and practically is the formation of different flavors of the complex of the same arrestin–receptor pair. This might be determined by the positions of receptor-attached phosphates [54], but the ensemble of structurally distinct complexes likely exists even when arrestin binds a population of homogeneously phosphorylated receptors. It appears likely that distinct “active” conformations of bound arrestin have different signaling capabilities. Preliminary reports indicate the potential of biased ligands differentiating between arrestin isoforms, leading to different cellular events [148] or inducing distinct arrestin conformations, resulting in different biological consequences [163]. Thus, our understanding of the molecular mechanisms involved can pave the way to channeling arrestin-mediated signaling in desired directions, or away from undesired ones, by construction of arrestin isoform- or conformation-biased ligands. Enforcing a desired bias of arrestin-mediated signaling for research and therapeutic purposes has as much potential as current attempts to bias GPCR signaling to favor a particular G protein or to selectively channel it towards G proteins or arrestins [88]. This research avenue remains largely unexplored.

## Figures and Tables

**Figure 1 biomolecules-11-00218-f001:**
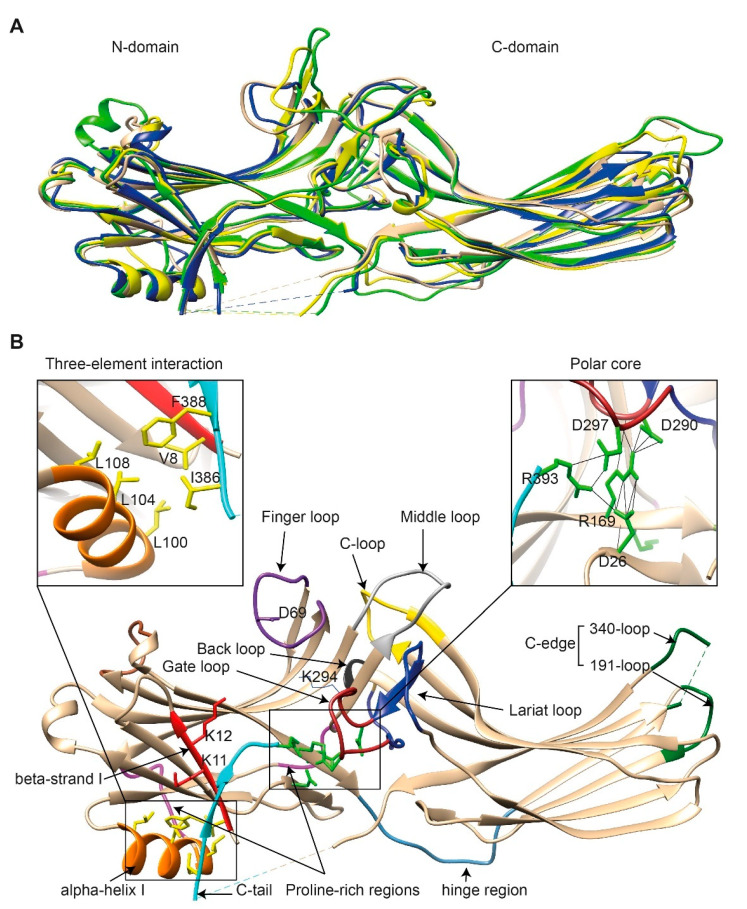
**The basal structures of arrestins.** (**A**) The crystal structures of bovine arrestin-1 (PDB: 1CF1, green [16]), bovine arrestin-2 (PDB: 1G4M, tan [17]), bovine arrestin-3 (PDB: 3P2D, blue [26]), and tiger salamander arrestin-4 (PDB: 1SUJ, yellow [20]) in the basal state are superimposed. Parts that are not resolved in the crystal structures are indicated by dashed lines. (**B**) Crystal structure of arrestin-2 (PDB: 1G4M [17]) in the basal state. Functionally important loops and critical residues are indicated and highlighted as follows: finger loop, purple; inter-domain hinge, blue-gray; β-strand I and the two lysines in it, red; α-helix I, orange; C-tail, light blue; hydrophobic side chains of residues in the α-helix, β-strand I, and β-strand XX of the C-tail mediating the three-element interaction, yellow; charged side chains of the five residues forming the polar core, green; polyproline motifs, light magenta; lariat loop, dark blue; its part called the gate loop, dark red; C-loop, yellow; back loop, black; C-edge loops, dark green. The side chain of the K294 in the gate loop pointing to the cavity of the N-domain is also shown. Close-up views of the three-element interaction and the polar core are shown in the left and right inserts, respectively.

**Figure 2 biomolecules-11-00218-f002:**
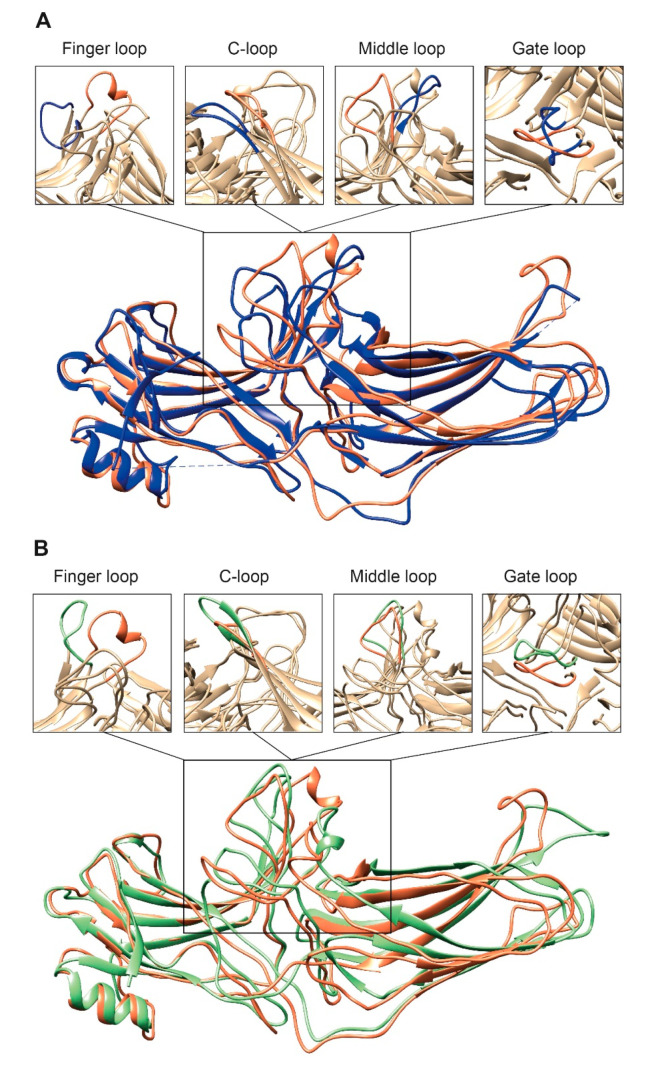
**Comparison of arrestin-2 conformations in complex with M_2_R and NTS_1_R.** (**A**) The crystal structures of arrestin-2 in the basal state (PDB: 1G4M, blue [17]) and in complex with NTS_1_R (PDB: 6UP7, orange [34]) are superimposed. (**B**) The crystal structures of arrestin-2 in complex with NTS_1_R (PDB: 6UP7, orange [34]) and in complex M_2_R (PDB: 6U1N, green [33]) are superimposed. The conformations of critical arrestin elements participating in receptor binding are compared in the close-up views above each panel. Specific regions are selected for comparison and the rest of structures are colored tan to highlight the difference in rearrangements upon activation.

**Figure 3 biomolecules-11-00218-f003:**
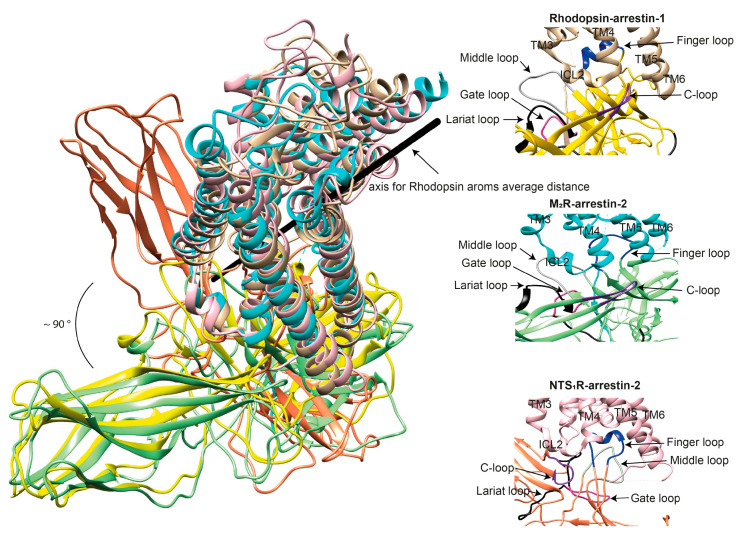
**Comparison of the conformations of arrestin-1 and arrestin-2 in complex with different receptors.** (**Left panel**) The structures of the rhodopsin-arrestin-1 (PDB: 5W0P [43]), M_2_R-arrestin-2 (PDB: 6U1N [33]), and NTS_1_R-arrestin-2 (PDB: 6UP7 [34]) complexes are superimposed according to the best match in receptor sequences, which shows about a 90° difference in the position of arrestin-2 in complex with NTS_1_R as compared to the other structures. (**Right panels**) Close-up views of the central crests of arrestins in contact with the receptor are shown for the rhodopsin (tan)-arrestin-1 (yellow) interface in the upper panel, the M_2_R (cyan)-arrestin-2 (green) interface in the middle panel, and the NTS_1_R (pink)-arrestin-2 (orange) interface in the lower panel. The finger (blue), middle (gray), lariat (black), gate (violet), and C- (purple) loops are highlighted to show the difference in the configurations of these elements in the structures.

**Figure 4 biomolecules-11-00218-f004:**
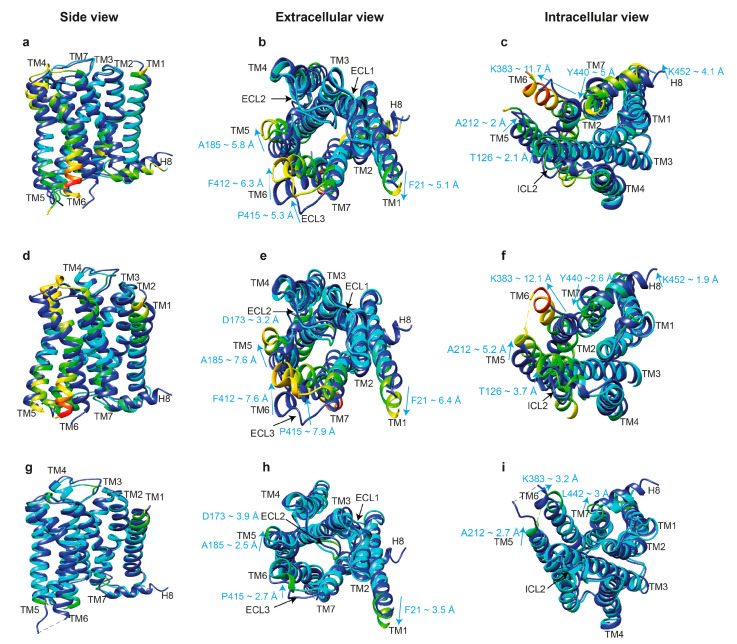
**Comparison of the crystal structures of inactive and active M_2_R.** (**Upper panels**) The structures of inactive M_2_R (PDB: 3UON, blue [73]) and active M_2_R-G_oA_ complexes (PDB: 6OIK, rainbow [74]) are superimposed in side view (**a**), extracellular view (**b**), and intracellular view (**c**). Residues of the active M_2_R-G_oA_ complex are color coded based on the RMSD of α-carbons from the matched residues in the inactive receptor with thresholds of 1.5 (cyan), 2.5 (green), 5 (yellow), 7.5 (orange), and 12 Å (red) to show the differences between the structures. The direction and extent of movement of residues with significant differences between these structures are shown with cyan arrows in Å. (**Middle panels**) The structures of inactive M_2_R (PDB: 3UON, blue [73]) and active M_2_R-arrestin-2 complexes (PDB: 6U1N, rainbow [33]) are superimposed in side view (**d**), extracellular view (**e**), and intracellular view (**f**), and segments in the arrestin-2 bound structure are color coded based on the RMSD of α-carbons as described for the upper panel. (**Lower panels**) The structures of active M_2_R-G_oA_ (PDB: 6OIK, blue [74]) and M_2_R-arrestin2 complexes (PDB: 6U1N, rainbow [33]) are superimposed in side view (**g**), extracellular view (**h**), and intracellular view (**i**), and segments in the arrestin-2 bound structure are color coded based on the RMSD of α-carbons as described for the upper panel.

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
