# Peer review of "Receptor-Arrestin Interactions: The GPCR Perspective"

_biomolecules, 2021, doi:10.3390/biom11020218_

Round 1
Reviewer 1 Report
See attached file.

Author Response
While the review is not focussing on arrestin-mediated signalling, it is mentioned several times. What is completely missing from the paper is recent evidence that arrestin-mediated signalling always has a G protein component. Arrestins certainly have a scaffolding function in several signalling cascades (the review correctly mentions that the JNK3 cascade is the one that is best investigated) and therefore signalling through these cascades is disrupted if arrestins are knocked down. However, it has now been convincingly demonstrated that arrestins do not catalyze signals without any G-protein involvement (key reference: (1) but see also (2)). This has to be mentioned somewhere.
The reviewer is correct. While the review focuses on the molecular mechanism of arrestin binding to GPCRs, it is important to comment on arrestin-mediated signaling alluded to in several places. We added a paragraph on that issue in section 9.
The completion of the phosphorylation code of the beta2-adrenergic receptor (discussed on page 16 of the review) was first described in reference (3).
Thanks! Appropriate reference added.
The finding that other kinases than GRKs can promote arrestin binding (discussed on page 16) is
much older than the reference given by the authors (ref132), see for example reference (4).
The barcode hypothesis was first proposed by Andrew Tobin (he did not call it „barcode hypothesis“, though), even though one gets a different impression when looking at the literature. The reference is even used in the review (ref 147) but it should be credited in line 693 as well.
Thank you! Missing reference added and this part of the review was re-written.
Minor issues in the text
line 47: „it is important from a biological standpoint“
line 173: „superimposable“
line 253: „movement of Y44 in the conserved NPxxY motif“
line 261: „D120 in the DRY motif“
line 263: „in the NPxxY motif“
line 397: „Based on ... evidence, the arrestin finger loop ...“
line 401: „In fact, the finger loop ...“
Thanks! These and other issues with the use of “a” and “the” were corrected throughout the manuscript by a native speaker of American English.
Lots of issues in the References section
The authors should cite all journals in an abbreviated consistent manner. For some reason, this
concerns particularly papers from the J Biol Chem, which is often quoted as „The journal of biological
chemistry“ (references 25, 28, 29, 32, 33, 34, 41, 58, 61, 63, 64, 81, 95, 98, 99, 110, 111, 117, 122,
125, 127, 130, 131, 132, 133, 134, 135, 136, 139, 140, 141, 143, 144, 152, 153, 154). But there are
also others:
„Trends in biochemical sciences“ should be „Trends Biochem Sci“ (reference 14)
„Cell research“ should be „Cell Res“ (references 36, 73)
„Scientific reports“ should be „Sci Rep“ (reference 48, 155)
„Cellular signalling“ should be „Cell Signal“ (reference 59)
„Nature Structural & Molecular Biology“ should be „Nat Struct Mol Biol“ (reference 60)
„Nature Chemical Biology“ should be „Nat Chem Biol“ (reference 83)
„Journal of the American Chemical Society“ should be „J Am Chem Soc“ (reference 88)
„Nature Communications“ should be „Nat Commun“ (reference 94)
„Proceedings of the National Academy of Sciences“ should be „Proc Natl Acad Sci U S A“ (reference
116)
„Trends in Pharmacological Sciences“ should be „Trends Pharmacol Sci“ (reference 147)
„The Journal of Neuroscience“ should be „J Neurosci“ (reference 150)
„Science Signaling“ should be „Sci Signal“ (reference 158).
The journal style seems to be not to use dots for abbreviation, so „J. Biol. Chem.“ is also incorrect and
should be converted to „J Biol Chem“ (references 27, 62, 100, 119) and „J. Mol. Biol.“ to „J Mol Biol“
(reference 42).
The authors should take great care with capitalization. I believe this to be a problem of the
bibliography program used but then you have to manually fix your References section. For example,
„gpcr“ should be converted to „GPCR“, „erk“ to „ERK“, „grk“ to „GRK“, „mapk“ to „MAPK“, „jnk3“ to
„JNK3“, „g protein“ to „G protein“, „go“ to „Go“, „cxcr7“ to „CXCR7“, „class b“ to „class B“, „arg“ to
„Arg“, „cryo-em“ to „cryo-EM“, „dry“ (for the motif) to „DRY“, „at1“ to „AT1“, „kda“ to „kDa“, „ip6“
to „IP6“, „nmr“ to „NMR“, „angiotensin ii“ to „angiotensin II“ etc.
For the structural papers (References 20, 21, 24, 31) make sure that the Ångstrom sign is correctly
represented as Å (i.e. neither „a“ nor „A“ nor „å“).
Betas are sometimes mistyped as „b“ (e.g. reference 27, 117).
Reference 22 = Reference 23.
Reference 26 is cited as „in press“ even though it was published in 2007.
References 50 and 145 are incomplete.
Reference 128 is only referred with a DOI. The correct reference is Elife 4: e05981.
Thanks! Many of these errors were generated by EndNote. We corrected the errors in this section.
Reviewer 2 Report
The authors highlight interesting structural insights into arrestin-GPCR complexes, by decomposing how proteins interact, in order to determine if common features can be isolated between all partners. They address many specific issues and draw the attention to the structural consequences of phosphorylations in binding recognition and selection. The manuscript is however far unbalanced with paragraphs containing well-structured sentences and ideas, and many others very unbalanced. Some remarks follow.
The introduction is rather vague and badly structured, the authors need to address more specifically what they are willing to explain, and avoid sentences such as “Fine molecular details might be of interest only to those who work on structure-function of arrestins and GPCRs.” (lines 54-55) If this article is meant to be a review, then we expect to get those “details”.
The authors should also order their arguments, some sentences seem repeated without any order (I think I got the point that there are 4 arrestin types). Line 65, is it “arrestin-1 a” or “arrestin-1 alpha”?
Line 125: what is the resolution of the crystal structure?
Why do you quote “legs” and not detail “lariat”? (and do not quote it ?) Do not quote any, define each term once, and refer to it afterwards … If a term is not precise enough, change it.
Why is figure 4 indicated before figures 2 and 3? Figures need to be renumbered according to the text appearance.
Section 1,2 and 3 need a large rewriting to be more consistent with the rest of the manuscript.
Lines 234 and 2336 “GPCR activation is accompanied by the movement of transmembrane (TM) helices”: which are?
Line 255, please precise what the (TMx).50 number is for… Why 50?
In Figure 2, some TM labels are superimposed on the cartoon representations, they should be offset to increase readability.
Lines 297-298: “Additionally, let’s compare how much of the cytoplasmic side of the receptor do G
proteins and arrestins engage.” Be my friend… Update this sentence to have a more scientific vocabulary.
Lines 342 to 360: what that not stated above already?
Line 435, “does the cell “know” which particular receptor arrestin is bound to.”. Please don’t mix knowledge (cell contains) and human-centric views (“cell knows”).
In the references section, some corrections should be performed: GPCR and not gpcr, G-proteins and not g-proteins, etc.
References 23 and 23 are duplicate.
The authors have also biased a lot their bibliography with a mere one third of self-citations, while in the meantime not citing a prolific GPCR+arrestin author like K. Palcezwski (to which I am not affiliated at all) or other reference authors in the filed (GPCR structural consortium, some Novel prizes for instance)… This is not good practice.
In short, this article needs a major overhaul in some sections to be published.
Author Response
The introduction is rather vague and badly structured, the authors need to address more specifically what they are willing to explain, and avoid sentences such as “Fine molecular details might be of interest only to those who work on structure-function of arrestins and GPCRs.” (lines 54-55) If this article is meant to be a review, then we expect to get those “details”.
The intro was modified and indicated sentence was re-written. Still, we wanted to acknowledge the fact that a lot more people are interested in what is actually happening than in what particular lysine or asparagine does the job.
The authors should also order their arguments, some sentences seem repeated without any order (I think I got the point that there are 4 arrestin types). Line 65, is it “arrestin-1 a” or “arrestin-1 alpha”?
In part this issue arose because we attempted to follow the structural details in referenced papers and explain to non-structural biologists why do we need to know those details. That “a” is superscript, referring to a footnote.
Line 125: what is the resolution of the crystal structure?
Thanks! The resolution is now specified.
Why do you quote “legs” and not detail “lariat”? (and do not quote it ?) Do not quote any, define each term once, and refer to it afterwards … If a term is not precise enough, change it.
The term “leg” replaced with “connector”. Lariat loop defined and quoted several times.
Why is figure 4 indicated before figures 2 and 3? Figures need to be renumbered according to the text appearance.
Thanks! Figures are renumbered now.
Section 1,2 and 3 need a large rewriting to be more consistent with the rest of the manuscript.
The sections were modified.
Lines 234 and 2336 “GPCR activation is accompanied by the movement of transmembrane (TM) helices”: which are?
We specified that TM6 moves more than other TMs.
Line 255, please precise what the (TMx).50 number is for… Why 50?
Thanks for pointing this out. We fully explained Ballesteros-Weinstein numbering system in revision. Part of it is that the most conserved residue in each TM was arbitrarily assigned number 50. Hence “large” numbers.
In Figure 2, some TM labels are superimposed on the cartoon representations, they should be offset to increase readability.
Thanks! The labels were moved.
Lines 297-298: “Additionally, let’s compare how much of the cytoplasmic side of the receptor do G
proteins and arrestins engage.” Be my friend… Update this sentence to have a more scientific vocabulary.
We rephrased this sentence.
Lines 342 to 360: what that not stated above already?
We don’t quite get what the problem is. We did not state this explicitly before.
Line 435, “does the cell “know” which particular receptor arrestin is bound to.”. Please don’t mix knowledge (cell contains) and human-centric views (“cell knows”).
We rephrased this sentence.
In the references section, some corrections should be performed: GPCR and not gpcr, G-proteins and not g-proteins, etc.
References 23 and 23 are duplicate.
Thanks! Many of these errors were generated by EndNote. We corrected this section.
The authors have also biased a lot their bibliography with a mere one third of self-citations, while in the meantime not citing a prolific GPCR+arrestin author like K. Palcezwski (to which I am not affiliated at all) or other reference authors in the filed (GPCR structural consortium, some Novel prizes for instance)… This is not good practice.
We cited papers based on their content and relevance to the subject discussed, not on authorship. As far as Dr. Palczewski is concerned, his group did not publish papers directly related to arrestins since 2006. We cited his seminal discoveries of the detachment of arrestin C-terminus upon receptor binding and of the mechanism of GRK activation by its binding to active GPCR, as well as his associates’ remarkable prediction of the arrestin domain twist years before any structures of GPCR-bound arrestins were solved. To the best of our knowledge, there were only two Nobel prize winners in this field, Drs. Lefkowitz and Kobilka, which were cited 19 and 13 times, respectively.
In short, this article needs a major overhaul in some sections to be published.
Done.
Reviewer 3 Report
The review by Mohamed Seyedabadi et al. entitled “Receptor-arrestin interactions: the GPCR perspective” is a timely and inspiring review that covers a neglected part in GPCR research. I enjoyed reading it a lot and I only have a few minor comments listed below.
1) I thank the authors for using the term “activation” or “active” arrestin in parenthesis since we currently don´t know what this really means structurally. However, the parenthesis are not used consistently throughout the manuscript. Either this has a meaning or was forgotten.
“ “ used in line 37, 39, 54, 554, 750 forgotten line 116? Please check…
2) line 80: double space between …might be…
line 428: double space between …loop was…
3) page 11 line 396 to 400: please check sentence for consistency:
…to the alpha5 helix of the Ras domain in G in the GPCR…
Is there something missing after G?
4) Figures are not numbered consistently: Order of figures is 1,4,3,2
Change the numbering of 4 and 2 and check throughout the manuscript when figures are mentioned.
5) In my opinion figure 3 is a bit difficult for the reader to keep oriented. Please consider to add an axis through the receptor for orientation, particularly since the orange arrestin appears to be at the same height as the receptor.
6) Please check the references mentioning none GPCRs as target for arrestin for correctness (Ref. 7 to 12). At least Ref 8, 10, 11, and 12 don’t look familiar to me and do not contain the word arrestin when you search the document for arrestin using acrobat reader… Other references would suite better: eg PMID: 29776602, PMID: 19888962
Author Response
The review by Mohamed Seyedabadi et al. entitled “Receptor-arrestin interactions: the GPCR perspective” is a timely and inspiring review that covers a neglected part in GPCR research. I enjoyed reading it a lot and I only have a few minor comments listed below.
1) I thank the authors for using the term “activation” or “active” arrestin in parenthesis since we currently don´t know what this really means structurally. However, the parenthesis are not used consistently throughout the manuscript. Either this has a meaning or was forgotten.
“ “ used in line 37, 39, 54, 554, 750 forgotten line 116? Please check…
Thank you! Corrected.
2) line 80: double space between …might be…
line 428: double space between …loop was…
Thank you! Corrected.
3) page 11 line 396 to 400: please check sentence for consistency:
…to the alpha5 helix of the Ras domain in G in the GPCR…
Is there something missing after G?
Thanks! Rephrased.
4) Figures are not numbered consistently: Order of figures is 1,4,3,2
Change the numbering of 4 and 2 and check throughout the manuscript when figures are mentioned.
Thanks! Figures renumbered.
5) In my opinion figure 3 is a bit difficult for the reader to keep oriented. Please consider to add an axis through the receptor for orientation, particularly since the orange arrestin appears to be at the same height as the receptor.
Thanks! The axis for rhodopsin in the complex was added in Fig. 3.
6) Please check the references mentioning none GPCRs as target for arrestin for correctness (Ref. 7 to 12). At least Ref 8, 10, 11, and 12 don’t look familiar to me and do not contain the word arrestin when you search the document for arrestin using acrobat reader… Other references would suite better: eg PMID: 29776602, PMID: 19888962
Thanks! References corrected.
Reviewer 4 Report
This is an extremely comprehensive review of the interaction of arrestins with GPCRs. The authors describe in detail the interactions that occur upon GPCR-arrestin binding, and discuss the different forms this interaction can take. They review the results of many different studies to detail the complexity and the variability of arrestin-GPCR interactions. This is a well written review and I do not have many criticisms, with my only general comment being that although the English is clear and understandable, it also seems likely that it was not written by a native speaker, as there are many instances where words like “the, a, an” etc have been omitted. Thus, while editing of the English is not necessary for clarity, it would improve its general readability. Other than that, I only have a few additional minor comments.
- Some of the figures are in the wrong order. The Figures go 1, 4, 3, 2, they should be 1, 2, 3, 4.
- There are many instances in the paper where there are clearly two or more spaces between words/sentences where it should only be one.
- Line 398: this sentence is unclear: “similar to the α5 helix of the Ras domain in G in the GPCR-G protein complexes”
Author Response
This is an extremely comprehensive review of the interaction of arrestins with GPCRs. The authors describe in detail the interactions that occur upon GPCR-arrestin binding, and discuss the different forms this interaction can take. They review the results of many different studies to detail the complexity and the variability of arrestin-GPCR interactions. This is a well written review and I do not have many criticisms, with my only general comment being that although the English is clear and understandable, it also seems likely that it was not written by a native speaker, as there are many instances where words like “the, a, an” etc have been omitted. Thus, while editing of the English is not necessary for clarity, it would improve its general readability. Other than that, I only have a few additional minor comments.
Thanks! We carefully edited the manuscript with the help of a native speaker of American English to put all articles in place.
- Some of the figures are in the wrong order. The Figures go 1, 4, 3, 2, they should be 1, 2, 3, 4.
Thanks! The figures were renumbered.
- There are many instances in the paper where there are clearly two or more spaces between words/sentences where it should only be one.
Thanks! Extra spaces eliminated. They were actually introduced by MDPI.
- Line 398: this sentence is unclear: “similar to the α5 helix of the Ras domain in Ga in the GPCR-G protein complexes”
Thanks! Corrected.